# A Study Evaluating Consumer Motivations, Perceptions, and Responses to Direct-to-Consumer Canine Genetic Test Results

**DOI:** 10.3390/ani12233360

**Published:** 2022-11-30

**Authors:** Nikki E. Bennett, Peter B. Gray

**Affiliations:** Department of Anthropology, University of Nevada, Las Vegas, NV 89154, USA

**Keywords:** *Canis lupus familiaris*, human-dog bond, direct-to-consumer model, personal genomic services, anthrozoology, identity construction, breed options theory

## Abstract

**Simple Summary:**

Numerous studies have evaluated the human personal genomics industry, though research has largely overlooked consumer use of these tests for companion animals. This is surprising considering the domestic dog and cat genomes were sequenced shortly after the completion of the human genome and the first genetic test became available for dogs in 2007. As a novel area of research, this study draws on a previous analysis of companion animal genetic company website and consumer review data. The theoretical framework builds on human–animal studies and research into human self-use of personal genomic services. Our objectives were to evaluate consumer (1) motivations for having their dog genetically tested, (2) perceptions of the returned results and services, and (3) responses to their dog’s results. Results showed that dog guardians are motivated to use these services to learn their dog’s breed composition, perceive the results as accurate, and respond by sharing their dog’s results with family and friends. Using these results and the presented framework, future research is positioned to evaluate how consumers may selectively incorporate genetic test results into their relationship with their dog.

**Abstract:**

Direct-to-consumer genetic services allow companion animal guardians to purchase a DNA test and receive detailed results about their pet’s ancestry, health, and traits results. In collaboration with Wisdom Panel, we present novel findings about consumer motivations, perceptions, and responses to their use of canine genomic services. Wisdom Panel customers were invited to complete an online survey anonymously in which they were asked about their reasons for using a genetic test for their dog, how they perceived the test’s results, and how they responded to the results they received. Participant data revealed most utilized a test that provided more ancestry/breed results (75.9%) as compared to health-related results. The majority of participants perceived the breed test results as accurate (52.0% strongly agree, 27.6% somewhat agree) and the genetic services provided as having great value (49.6% strongly agree, 32.7% somewhat agreed). In responding to their dog’s results, participants indicated they shared the information with family (88.1%) and friends (84.2%). Collectively, our study indicates consumers are more focused on their dog’s ancestry than other test results. Using these findings and previous literature on human direct-to-consumer genetic testing, human–animal dyads, and identity construction, we consider the possibility of “breed options theory” and future areas of research.

## 1. Introduction

The direct-to-consumer (DTC) genomic market–both human and nonhuman–allows consumers to purchase various tests that can provide ancestry and health-related results, even to the extent of having whole genomes sequenced. Recent market analysis from Global Market Insights Inc. predicts the human genetic test market will be worth $31.8 billion [1] and the animal genetics market $6.4 billion [2] globally by 2027. The DTC pet genetic industry has received little recognition in scholarly literature until recently. This is surprising considering the domestic dog and cat genomes were sequenced in 2004 and 2007, respectively [3,4], in addition to the first dog genetic test (GT) becoming available in 2007 [5].

Mars Petcare provided the first dog DNA test that initially required a veterinary care provider to collect a blood sample for analysis [5]. Within 2 years of its launch, Mars Petcare rebranded their DNA tests as Wisdom Panel^TM^ and moved to a DTC relationship, allowing companion animal guardians (i.e., the primary caregiver of an animal that may or may not be viewed as a “pet”) to purchase tests, collect a cheek swab (or cheek cell sample) from the animal, and receive detailed results without the need of a veterinarian. In addition to Wisdom Panel’s services, other companies have entered the market (e.g., Embark), with Basepaws launching the first DTC cat DNA test in 2016 [6]. Despite this industry’s growth and popularity in which consumers have numerous options for getting a nonhuman animal’s genome sequenced, research from this industry has been limited to the scope of population genetics and other genotype-phenotype related outcomes.

Dog genetic information derived from DTC genetic tests (DTC-GT) has provided a wealth of information to the researchers collaborating with these companies. For example, Deane-Coe et al. [7] collaborated with a DTC pet genomic company to study canid trait ancestries and breed populations. In doing so, they were able to detect low frequency alleles associated with regional and functional selection within breeds; develop visual models to illustrate historical breed connections; and compare subjective breed assessments to objective genetic data. Another study by Lewis and Mellersh [8] empirically evaluated the efficacy of DTC-GTs–something that has been called into question (e.g., [9])–to determine if allele frequency for autosomal recessive diseases changed following DNA tests becoming commercially available. Their results found meaningful declines in harmful mutation frequencies in UK Kennel Club registered dog populations, implying DTC-GTs can have positive benefits on animal health when incorporated into breeding practices. This trend is particularly salient to purebred dog populations as they are more likely to have health problems. Likewise, Bannasch et al. [10] used genomic data derived from DTC tests to make comparisons between breed standards and breed-specific health outcomes. Subsequently, they found associations between certain dog breeds (e.g., brachycephalic dogs) having higher morbidity rates and higher levels of inbreeding. Clearly, the use of animal genomic data resulting from DTC genetic companies has significantly contributed to our understanding of genomics and associated topics. However, research has largely neglected consumer perspectives on the DTC animal genomic market.

Social science research into human self-use of DTC-GTs has been essential to improving the industry by providing insight into consumer motivations to use genetic services, their understandings of test results, and behavioral responses to their genetic test results (e.g., [11,12,13]). In a unique content exploration of the DTC-GT pet industry, Bennett et al. [14] reported on company website and consumer review content. They found DTC pet animal genetic companies primarily advertised products and services, with themes related to the company’s credentials and trustworthiness to establish the merit of returned test results. Of the products and services marketed, genetic companies promoted health-related tests the most, with breed results as their second focus. Consumer reviews provided on Amazon were generally positive, with consumers principally sharing their perceptions about test accuracy, reasons for pursuing (e.g., to learn their pet’s breed), and experience using the tests (e.g., ease of use). Aside from this study, the remaining literature surrounding the DTC market does so from a theoretical or commentary perspective (e.g., [9]). 

To overcome the limitations of passive data use (i.e., Amazon Reviews, company website data), we collaborated with Wisdom Panel^TM^ to recruit participants from their existing consumer database. By collecting data directly from consumers, our goal was to further evaluate their motivations, perceptions, and responses to their using DTC DNA services for their dog. These areas of research are important to understand consumer experiences and potential implications to the human-dog bond. As such, our objectives were to: Evaluate consumer motivations to pursue genetic services for their dog.Gain insight into consumer perceptions about genetic test services and their dog’s associated results.Assess consumer responses after receiving their dog’s genetic test results to include sharing the results with others, resources consulted, and incorporating the knowledge gained from the results.Collectively use study results and assess methodologies to develop future project designs for larger scale data collection about consumer use of pet animal genetic services.

## 2. Materials and Methods

This research was approved by the University of Nevada, Las Vegas’ Internal Review Board (protocol number 2021-122). Wisdom Panel collaborators generated a randomly selected sample that consisted of US consumers who used a Wisdom Panel Essential [15] or Premium [16] canine genetic test before July 2021, a time frame selected by Wisdom Panel staff to be reflective of the same user experience. Participants were recruited via email in February 2022 in which two emails (11 and 19 February) went out to approximately 80,000 Wisdom Panel customers. This large number of potential participants was based on Wisdom Panel market insights anticipating a less than 1% response rate. The survey link was closed on 1 March 2022 as no responses had been collected for 48-h. To ensure participant privacy, UNLV researchers did not have access to the selected consumer base and no identifying information was collected during the survey. Participation required individuals to be at least 18 years old and participants were not compensated for participating. 

### 2.1. Survey Measures and Procedures

The following survey sections covered dog guardian motivations to pursue genetic testing for their dog, perceptions regarding the genetic services and test, and responses upon receiving their dog’s genetic test results. Here, we operationalize “dog guardian” as an all-encompassing title to be inclusive of the various views one may have about a dog to which they are the primary caregiver. For example, dog breeders may not view certain dogs as a companion and people who have a companion dog may view their dog as family or only as a “pet” (e.g., [17]).

Both quantitative (nominal, ordinal) and qualitative (open-ended text entries) data were collected. The survey was administered online using Qualtrics [18], opened with the participant consent form, and proceeded for those who agreed to participate. The next question asked, “Have you ever used a Wisdom Panel genetic test for a dog?” to screen participants. Participants who indicated they had not previously used a Wisdom Panel dog genetic test were directed to the end of the survey and those who answered “yes” proceeded through the remainder of the survey. Refer to Appendix A for complete survey questions.

The first survey section asked about general experiences to include: the specific type of Wisdom Panel test used, their means of acquiring the test (e.g., purchased online), and how they learned about Wisdom Panel services. Next, a question asked for the name of the dog they had genetically tested to incorporate Qualtrics’ piped text feature in which questions would use the dog’s name into statements and responses (e.g., “If I had a question about [dog’s name]’s test results, I would…). Participants who had more than one dog tested were directed to focus on their most recent experience, but if they felt strongly about completing the survey for more than one dog they could retake the survey for their other dog(s). Because no identifying information was collected from participants, we are unable to report if any completed the survey more than once. 

The next set of questions asked participants to provide their dog’s demographics (e.g., age, spay/neuter status). From there, the following section assessed participant motivations to pursue a DNA test for their dog. These questions were adapted from Carere and colleague’s [19] study that surveyed 23andMe consumers and Bennett et al.’s [14] qualitative study on the companion animal genetic industry. The next question set evaluated the participant’s perceptions about their dog’s genetic test results and were also created from Bennett et al.’s [14] study in which consumers shared several themes related to the perceived utility of the genetic test. Both sections–motivations and perceptions–used a 5-point Likert scale for participants to rate their agreement with questionnaire items (Strongly Agree, Somewhat Agree, Neither Agree nor Disagree, Somewhat Disagree, Strongly Disagree).

The following survey portion asked participants about their responses to their dog’s genetic test results. This section incorporated Qualtrics’ skip logic to best target questions to participants based on their behavior after receiving their dog’s results. The first question asked participants if they had discussed their dog’s genetic results with anyone. Participants who selected “yes, [they] discussed the results with someone,” were then asked who they discussed the results with and what aspects of the results they discussed. Participants who selected “no, [they] did not discuss the results with anyone,” were asked why they did not share their dog’s results with others. Additional behavioral responses evaluated were related to consumers making care decisions after receiving their dog’s test results. Collectively, these questions were also adapted from Carere et al. [19] and Bennett et al.’s [14] study that detected DTC genetic test user patterns in which they respond to test results in various ways to include talking with someone about the results and altering their own or their animal’s care. The final section collected participant demographic information related to self-identified gender, ethnicity, income, and education. Survey questions are provided as a Appendix A.

### 2.2. Data Processing and Analysis

Data analysis was completed using SPSS version 28 and Microsoft Excel. Descriptive statistics were used to characterize all measures evaluated (dog and participant demographics, motivations to use the test, perceptions of the test and services used, and responses to test results). Qualitative responses provided in “Other (Please explain)” were evaluated to determine if (1) the participant provided a response that was a selection choice but did not select it or (2) the participant did have an “other” response. Qualitative responses that were determined to be a provided answer choice but the participant did not select the provided response, were grouped accordingly and removed from the “Other” category. For example, in answering the question “How did you learn about Wisdom Panel products,” some participants typed they learned about the test from someone else (e.g., friend) who had used the test. Considering the answer choice “Recommended to me by someone who had previously used a pet genetic test” was provided, this response was grouped as such. As it relates to qualitative responses revealing a pattern that was not provided as an answer choice, we report on them within the “Other” category as a sub-theme.

## 3. Results

### 3.1. Sample Description–Participant, Dog, & Test Type

Of the 301 participants who started the survey and responded to the informed consent question, 253 completed the survey in its entirety, reflecting an 84.1% completion rate and <1% potential participant recruitment. Partial responses were included in analyses and Figure 1 shows the survey’s logical order with the number of participants who completed each respective section.

Table 1 summarizes participant sociodemographics and Table 2 provides canine demographics. In sum, our sample consisted primarily of women (83.4%) who self-reported being white (88.1%), and educated (39.1% Bachelor’s degree, 15.8% Master’s degree, 15.0% some college).

Most participants reported having a dog that was a neutered male (48.3%) or spayed female (45.2%), and was categorized as a companion animal/family member (97.3%). Dog mean age was 4.48 years (min 0.25, max = 17.50, SD = 3.82), they had been in the participant’s care for an average of 3.07 years (min = 0.00, max = 16.00, SD = 3.37), and were acquired via adoption from an animal shelter or rescue organization (71.1%).

Table 3 provides a breakdown of the test type, method of purchasing, and how participants learned about Wisdom Panel canine genetic services. The majority of participants bought the Wisdom Panel Essential test (75.9%), which–at the time of this writing–is advertised to screen for 350 breeds, 35 traits (e.g., coat type), 25 medical complications (“Identify risks before your pup has medication or surgical procedures”), and trace dog ancestry up to the great-grandparents [16]. 55.7% of participants purchased the test directly from Wisdom Panel and 30.9% did so through a third party e-commerce website. Participants primarily learned about Wisdom Panel from someone who recommended the service (25.1% by someone who had previously used Wisdom Panel services, 4.8% by a veterinarian, and 9.3% from someone on social media; Table 3). Of interest, when reviewing “Other (please explain)” responses, 15.1% described doing personal research (e.g., a general Internet search) and selected Wisdom Panel based on those results. One explanation read,


*I was a scientist in a direct to consumer dna [sic] testing for humans and looked into something similar for my dog. Wisdom seemed of reasonable accuracy for the cost.*


### 3.2. Motivations to Have Dog Tested

When asked about their primary reason for having their dog genetically tested, 85.2% (*N* = 218) reportedly did so to learn their dog’s breed, 7.0% (*N* = 18) to learn their dog’s health results, the test was a gift (4.7%, *N* = 12), or for other reasons (*N* = 8) (Table 2). When looking at participant responses for “Other (please explain),” they described reasons that were of equal importance but because the question only allowed one response, they used the text box to list those reasons together. For example, one participant said, “It was a birthday gift from my siblings because they knew how much I wanted to know what breeds made up [my dog].” Another participant described all of the test results as important,


*[To] learn breed results, health results, find any relatives she may [have] and character traits for training and what her learning habits would be.*


When answering Likert scale items about their motivations to use a genetic test for their dog, participants strongly agreed the Wisdom Panel test allowed them to: learn about their dog’s background because they had limited information about their dog (73.0%), satisfy their curiosity about their dog’s breed composition (72.3%), and participate in something fun and entertaining (53.1%) (Table 4).

### 3.3. Perceptions towards Test Results & Services

As shown in Table 5, participants had positive perceptions and attitudes towards the genetic test results they received. The majority of respondents indicated the provided breed results (52.0% strongly agree, 27.6% somewhat agree) and trait results (40.2% strongly agree, 33.5% somewhat agree) were accurate. Though, when reporting on the health results, 46.1% of participants indicated they neither agreed nor disagreed with the statement about their perceived accuracy. Another indicator participants had positive attitudes about their dog’s test results was most strongly agreed with the statement “I would recommend other dog owners use the test I did” (65.4%). Furthermore, 49.6% strongly agreed and 32.7% somewhat agreed with the Likert item that the genetic test used had great value.

### 3.4. Responses to Test Results

When asked if they had discussed their dog’s genetic test results with someone, 91.3% (*N* = 231) selected yes and only 8.7% (*N* = 22) selected no. Table 6 summarizes the contexts of their sharing their dog’s test results with others. Most participants selected they discussed their dog’s genetic test results with family (88.1%), friends (84.2%), and a veterinarian not employed by Wisdom Panel (54.5%).

When asked about what features they discussed with others, participants primarily shared the breed results (*N* = 229, 90.5%), with only 27.7% of participants (*N* = 70) sharing the health results and 19.8% sharing their dog’s results about “trait carrier status” (*N* = 50). Participants who selected “Other” (*N* = 6), added they discussed their dog’s “relatives” (*N* = 4) and “behavior/personality traits” (*N* = 2).

As it relates to the small number of participants who reported they did not discuss their dog’s test results with anyone, 2% (*N* = 5) reported they had not discussed the results because they did not think anyone else would be interested in them, another 2% (*N* = 5) selected “I plan to discuss [dog’s name]’s results with friends and family but haven’t gotten around to it,” and 1.2% (*N* = 3) selected “I don’t feel that [dog’s name]’s results are important enough to share.” Of note, 2 participants (0.8%) shared in the “Other (please explain)” text box that they did discuss the results with other people.

Participants were asked, “If a question were to arise after receiving [their dog’s] genetic test results, who would you consult?” The majority of participants reported they would consult a veterinary care provider (*N* = 194, 76.7%) or a Wisdom Panel associate (*N* = 87, 34.4% customer service; *N* = 75, 29.6% veterinarian/genetic counselor). Table 7 provides a comprehensive list of all choices and participant selections.

The next set of consumer response questions assessed if and how participants incorporated their dog’s genetic test results into their dog’s lives (Table 8). 67.6% of participants (*N* = 171) indicated they did “nothing else” with the results, while others indicated they altered their dog’s training (*N* = 53, 20.9%) and care (*N* = 24, 9.5%).

## 4. Discussion

Consumer interests in their dog’s ancestry may be reflective of their using the results to formulate an identity for both them and their dog. Research into human use of personal genomic services has consistently shown ancestry results impact a consumer’s “cultural or personal identity” [20]. For example, Marcon et al. [21] reported consumers used genetic tests to “solidify a sense of self” and create a genetic identity. Roth and Ivemark [22] also described test use to form a “racial/ethnic identity” and called this incorporation of ancestry information “genetic options theory.” Genetic options theory posits that DTC-GT users evaluate their ancestry information and selectively integrate ancestries that are positively perceived into their social identity [22]. This means the consumer will “...embrace identities [that] provide psychological or social value and ignore” the results that do not [22].

To consider the applicability of genetic options theory to pet guardians, it is important to discuss how people form an identity both for and with their pets. As it relates to their motives to pursue canine genetic test services, in addition to breed information, participants were looking to learn their dog’s “background” since they had a limited history about their dog. This incomplete background likely stems from most of the dogs reported in our sample as being adopted. Consumer use of DTC genetic services to learn an animal’s background may mirror human DTC-GT use in which previous research reported adult adoptees used these tests to learn their “race” and “search for biological family” [12]. Another study revealed consumers also used their results to learn missing heritage information such as withheld knowledge about biological relatives (e.g., sealed adoption records) [23]. While test use to locate canine relatives was not a major motivator in our sample, this use was detected. Canine relative results were a fairly new test feature at the time of this study, but considering dog guardians are beginning to incorporate these results into their learning about their dog, follow-up research can implement questions to evaluate these reasons. Furthermore, pet acquisition statistics for the United States [24,25] indicate this trend of adopting pets will continue. Therefore, consumer patterns of using tests to learn about a companion animal will also likely persist and indicates the importance of understanding how canine identity–related to breed and biological relatives–impacts the consumer.

Previous works have evaluated how companion animals situate within a pet guardian’s social identity and how guardians may view their companion animals as subject (e.g., friend, family), object (e.g., status symbol), or interchange between these views [26]. Jyrinki [27] found guardians formulate an identity through their pets in which pets were both a “means to connect” to others and “status communicator.” Focusing on pets as a social conduit, Wood et al. [28] reported companion animals “...facilitate relationships from which [pet guardians] derive tangible forms of social support....” Relatedly, and returning to Jyrinki’s idea of pets as a “status communicator,” pets as a signal of one’s social standing suggests “...pet [guardians construct] an external picture of themselves and look for [a] socially visible identity” [28]. A dog’s visible identity can be considered its physical qualities (e.g., body size, coat color) and how these phenotypes are associated with particular breeds. The value of dog breed and appearance has been shown to be an important factor influencing the type of dog purchased or adopted [29]. Even further, dog breed and appearance can be more influential in dog acquisition as compared to other considerations [30,31]. This trend is also reflected in a dog’s length-of-stay at an animal shelter based on its assigned breed category, with those assigned a breed with a negative-stereotype as taking longer to adopt [32].

As this relates to our findings, nearly all of the participants reported they had shared their dog’s results with someone. More specifically, they had shared their dog’s breed results with family and friends. For example, one participant wrote “When walking [my dog] at the park, I am often asked what breed he is.” This interaction demonstrates both social connectedness and the participant conveying their dog’s “breed identity” to others. Using these findings and previous literature on human DTC genetic testing, human–animal dyads, and identity construction we consider the possibility of “breed options theory” in which canine genetic test consumers are affected by their dog’s results, selectively incorporate the ancestry information into theirs and their dog’s identity, and share this identity with others. As to what aspects of a dog’s ancestry information are selectively incorporated, one may consider how culture influences breed stereotypes.

Dog breeds are often associated with specific behaviors, with assumptions regarding breed aggression as having societal implications. Breed Specific Legislation varies between U.S. cities and states, but most target dog breeds that are believed to pose a threat to the public [33]. Since BSL’s introduction, extensive research has been conducted to evaluate the effectiveness of these laws (e.g., [34]) and if breed-specific behaviors exist (e.g., [35]) (for a complete review on BSL, see [36]]. As this relates to breed options theory, those who confirm their dog’s breed composition through genetic services may selectively share their dog’s breed results based on what information is socially acceptable. As such, their selections may be influenced by breed stereotypes and the consumer’s expectations regarding their dog’s ancestry results. Future research should evaluate possible associations between dog guardian result expectation (i.e., guardian already has an established knowledge about their dog’s ancestry) and if their exposure to breed stereotyping impacts which ancestry information they share or apply to their dog’s breed identity.

Another consideration of consumer focus on breed results, as compared to health-related information, are those who view their dog as a family and/or companion member, as characterized by our sample. Participants reporting their dog as a companion/family member implies a shared positive relationship and, as such, maintaining that animal’s health may be perceived as more beneficial considering the shortened lifespan of domestic dogs in comparison to humans. These findings are highly relevant to the DTC-GT market as the pet animal industry markets health results over other test uses [14]. For example, Bennett et al. [14] reported companion animal genetic companies target “pet parents” by conveying the message that by using their services, “...pet owners... will know their pet better and pets will, in turn, live healthier and happier lives.” Despite company advertisement methods, consumers continue to be more focused on breed results. Societal value on pure-bred dogs over health outcomes is not a new phenomenon as research has shown breed popularity is more influenced by trends (“fads”) as compared to other considerations [30]. This is apparent in pop-cultural impacts in which movies featuring a dog “celebrity” stimulate interest in the breed of that particular dog [31].

### Limitations and Future Directions

As with all research, this study is subject to limitations. Our sample consisted primarily of participants who self-reported being female, white, educated, and viewed their dog as a companion/family member. We did not capture cat guardian use of genetic test services as our sample consists of only dog guardians. Our sample is also relatively small considering the popularity of this market and is from one industry-leading company [37,38]. Therefore, we are limited in being able to generalize consumer experiences to all demographics, namely those who do not identify their dog as a companion/family member, have used other companies (e.g., Embark, Basepaws), and used a test for a non-dog species. However, our objective was not to generalize across all consumers, but develop insights into an understudied market and offer ideas for future research in this industry.

Focusing on the broader scope of this industry, it will be important for future work to incorporate the veterinary care field into this discussion to collect data about their current experiences with (or lack of) incorporating genomics into their patients’ care. Banfield Pet Hospital [39] advertises their incorporation of Wisdom Panel tests into packaged services by saying,


*Get extra insight on your dog’s health, behavior and training, and other needs with a test that detects over 350 canine breeds and screens for genetic disease mutations.*


Wisdom Panel is not the only pet genomic service entering veterinary clinics as Neogen [40], a genetic company that does not rely on the DTC model, offers an array of genetic services for a variety of species. Instead of selling tests to consumers, Neogen only accepts companion animal genetic samples from veterinarians or related professionals with the exception of a cat genetic test that can be purchased online due to a partnership between Neogen and the Cat Fanciers Association [41]. In addition to their narrower model, they are actively educating veterinarians about canine genetic test results contribution to patient healthcare [42]. Therefore, future studies should incorporate veterinary care provider perspectives of genomic services into their framework.

Our last consideration for future research is related to our discussion on “breed options theory.” Using our findings and the framework outlined, more research is needed to understand potential implications genetic test services may have on the human-bond. Specifically, research should be designed to assess dog guardians before and after their using a genetic test for a companion animal. Such methods are more likely to make determinations about the impacts breed results may have on identity (e.g., do dog guardians use results to form a socially acceptable breed identity?) and their relationship with their dog (e.g., do they use the results to alter the care/training provided to that particular dog?).

## 5. Conclusions

Understanding pet guardian motivations, perceptions, and reactions to genetic test results are essential to evaluate potential impacts (both direct and indirect) this industry may have. Direct implications to consider are those intended by the industry, such as improving animal health outcomes as previously found [10], while indirect impacts are those not specifically advertised by pet animal DTC-GT companies. Examples to consider are consumer responses to unwanted test results (e.g., negative breed stereotypes) or this technology being used for other purposes.

In stark contrast with the human sector, animal genetic research is able to dabble in what most would believe to be science fiction. Somatic Cell Nuclear Transfer (SCNT), also referred to as “cloning,” has been applied to the companion animal industry [43] in which pet guardians are able to “genome bank” their cherished pet’s genetic material and eventually use this to create a clone (e.g., Viagen [44]). At the beginning of 2022, an article was released by Input Magazine detailing certain social media influencers had either already done so or put plans in place to clone their pet who had also become a social media celebrity [45].

Another application worth considering is the role animal DNA databases and SCNT may have in resurrecting extinct species [46], specifically, extinct dog breeds. Historically, extinct dog breeds were brought back by selecting individuals who were believed to have descended from the extinct breed and/or shared similar phenotypes (e.g., see Worboys et al.’s [47] account for the recreation of the Irish Wolfhound). An example of a currently extinct dog breed is the Woolly dog, a breed in which the fur was used to make textiles [48]. Ancient DNA analyses have become more accurate and the paleogenomic record is beginning to document the evolution of domestic dogs (e.g., [49]). Considering western societal obsession with dog breeds, consumer behaviors, and advanced genomic technologies, could a demand for extinct dog breeds be supplied?

In closing, partnerships between researchers and industry will continue to be essential to understand pet guardian use of genetic services and implications to the human-animal bond. By collaborating with Wisdom Panel, we were able to directly assess the consumer experience as well as evaluate the feasibility of future projects. In doing so, we have revealed novel insights into canine DTC-GT customer reasons to pursue genetic testing for their dog, views about the genetic test and services, and responses to their dog’s results. These factors, as well as how other agencies are impacted by the nonhuman animal DTC-GT industry, will continue to be important areas of research focus.

## Figures and Tables

**Figure 1 animals-12-03360-f001:**
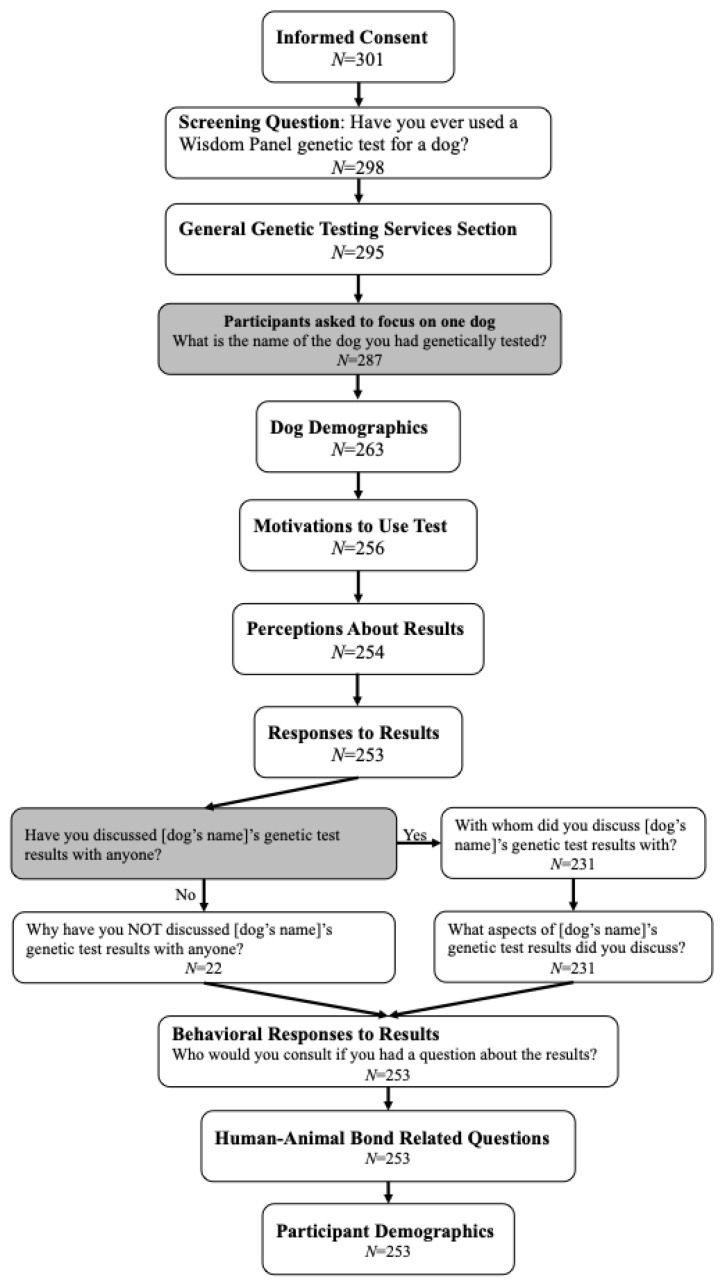
Survey flow with associated participant sample size. Survey sections with associated number of participants who completed each section. Gray boxes are questions that incorporated Qualtrics’ piped text (i.e., dog’s name) or display logic (i.e., Did they discuss the results with someone?).

**Table 1 animals-12-03360-t001:** Participant self-reported sociodemographic characteristics.

Demographic Variable	*N*	Percent
Sex		
Female	211	83.4
Male	34	13.4
Did not Disclose	8	3.2
Age (Years)		
18–25	5	2.0
26–35	25	9.9
36–45	36	14.2
46–55	66	26.1
56–65	55	21.7
65 and up	63	24.9
Did not Disclose	3	1.2
Racial/Ethnic Background		
White	223	88.1
Latino/Hispanic	13	5.1
Did not Disclose	13	5.1
Asian	2	0.8
American Indian/Alaska Native	1	0.4
Other	1	0.4
Annual Income (Household)		
Less than $30,000	7	2.8
$30,000–$59,999	30	11.9
$60,000–$89,999	40	15.8
$90,000–$129,999	43	17.0
$130,000–$149,999	19	7.5
$150,000–$199,999	24	9.5
More than $200,000	21	8.3
Prefer not to say	69	27.3
Education		
High school/GED equivalent	12	4.7
Some college	38	15.0
Associates degree	22	8.7
Technical or Vocational Training	7	2.8
Bachelor’s Degree	99	39.1
Master’s Degree	40	15.8
Doctorate or Professional Degree	19	7.5
Prefer not to say	16	6.3

**Table 2 animals-12-03360-t002:** Canine demographic variables as self-reported by participants.

Demographic Variable	*N*	Percent
Sex		
Neutered Male	127	48.3
Spayed Female	119	45.2
Intact Female	11	4.2
Intact Male	6	2.3
Relationship to Dog		
Companion/Family Member	256	97.3
Temporary Guardianship (Rescue, Fostered, Adopted to Someone Else)	3	1.1
Companion & Breeding Dog	2	0.8
Companion & Working Dog	1	0.4
Service Animal	1	0.4
Dog Acquisition		
Adopted from Animal Rescue or Shelter	187	71.1
Found as Stray	24	9.1
Other	18	6.8
Purchased from Breeder	15	5.7
Adopted from Someone Else Who Could No Longer Keep	10	3.8
Purchased from Online Ad	5	1.9
Purchased from a Pet Store	1	0.4
From Own Breeding Program	1	0.4
When Had Genetically Tested		
Within the Last Month	29	11.0
Within the Last Year	125	47.5
1 to 3 Years Ago	109	41.4
Primary Reason for Genetically Testing		
To Learn Breed Results	218	82.9
To Learn Health Results	16	6.1
Gift	12	4.6
Other	10	3.8
Dog Age & Time in Guardian’s Care		
	Mean ± SD	Min/Max
Age (Years)	4.48 ± 3.82	0.25/17.50
Length of Guardianship (Years)	3.07 ± 3.37	0.00/16.00

**Table 3 animals-12-03360-t003:** Description of test type, purchase method, and how participant learned about Wisdom Panel canine genetic services.

General Canine DNA Test Information	*N*	Percent
Test Type		
Essential (25+ health test results, ancestry/breed results, traits results)	221	75.9
Premium (Comprehensive health screening with 210 health test results, ancestry/breed results, traits results)	70	24.1
Purchase Method		
Purchased directly from Wisdom Panel webpage	162	55.7
Purchased from third party e-commerce website such as Amazon or Chewy.com	90	30.9
Was a gift to me from someone else	33	11.3
Other	6	2.1
Learned About Test (Select All that Apply)		
Recommended to me by someone who had previously used a pet genetic test	73	25.1
I saw a video commercial such as on television	18	6.2
I received an advertisement through email	13	4.5
I saw an advertisement on a social media platform such as Facebook or Reddit	66	22.7
Someone I follow and/or am connected with on social media shared a post recommending	27	9.3
It was a gift to me	31	10.7
My veterinarian recommended I use a genetic test	14	4.8
Other (Please explain):	-	-
Personal Research/Internet Search	44	15.1
Various responses with no Overarching Theme	27	9.3
Unsure/Can Not Recall	8	2.7
Recommendation by Third Party E-Commerce Platform	3	1.0

**Table 4 animals-12-03360-t004:** Participant responses to Likert items about their motivations for having their dog genetically tested. Reported as *N* (Percent).

The Genomic Information I Received from Wisdom Panel about [Dog’s Name] Allowed Me...	Strongly Agree	Somewhat Agree	Neither Agree Nor Disagree	Somewhat Disagree	Strongly Disagree
To satisfy my curiosity about [dog’s name]’s breed composition.	185 (72.3)	41 (16.0)	5 (2.0)	8 (3.1)	17 (6.6)
To see if [dog’s name] was at risk for a specific disease.	71 (27.7)	88 (34.4)	70 (27.3)	9 (3.5)	18 (7.0)
To learn about [dog’s name] without a veterinarian recommending the test.	77 (30.1)	32 (12.5)	95 (37.1)	10 (3.9)	42 (16.4)
To learn about [dog’s name] at the recommendation of my veterinarian.	5 (2.0)	7 (2.7)	86 (33.6)	22 (8.6)	136 (53.1)
To improve [dog’s name]’s health.	33 (12.9)	82 (31.6)	91 (35.5)	18 (7.0)	33 (12.9)
To find out if [dog’s name] is at risk of having an adverse response to some common medications.	47 (18.4)	67 (26.2)	84 (32.8)	20 (7.8)	38 (14.8)
To better plan for [dog’s name]’s future.	62 (24.2)	71 (27.7)	75 (29.3)	17 (6.6)	31 (12.1)
To participate in something fun and entertaining.	136 (53.1)	69 (27.0)	30 (11.7)	9 (3.5)	12 (4.7)
To satisfy my interest in genetics in general.	117 (45.7)	82 (32.0)	37 (14.5)	6 (2.3)	14 (5.5)
To participate in research.	33 (12.9)	45 (17.6)	112 (43.8)	27 (10.5)	39 (15.2)
To learn more about [dog’s name]’s background because I have limited information about [dog’s name].	187 (73.0)	42 (16.4)	14 (5.5)	2 (0.8)	11 (4.3)
To make inferences about [dog’s name]’s behavior.	95 (37.1)	84 (32.8)	49 (19.1)	12 (4.7)	16 (6.3)

**Table 5 animals-12-03360-t005:** Participant responses to Likert items about their perceptions, attitudes, and beliefs about the genetic test results received. Reported as *N* (Percent).

Questionnaire Item	Strongly Agree	Somewhat Agree	Neither Agree Nor Disagree	Somewhat Disagree	Strongly Disagree
[Dog’s name]’s breed results were accurate.	132 (52.0)	70 (27.6)	31 (12.2)	10 (3.9)	11 (4.3)
[Dog’s name]’s health results were accurate.	62 (24.4)	67 (26.4)	117 (46.1)	2 (0.8)	6 (2.4)
[Dog’s name]’s trait results were accurate.	102 (40.2)	85 (33.5)	50 (19.7)	8 (3.1)	9 (3.5)
Wisdom Panel genetic test results are trustworthy.	129 (50.8)	82 (32.3)	25 (9.8)	7 (2.8)	11 (4.3)
I would recommend other dog owner’s use the test I did for [dog’s name].	166 (65.4)	63 (24.8)	9 (3.5)	5 (2.0)	11 (4.3)
The Wisdom Panel genetic test I used for [dog’s name] has great value.	126 (49.6)	83 (32.7)	22 (8.7)	10 (3.9)	13 (5.1)

**Table 6 animals-12-03360-t006:** Participant data about who they discussed their dog’s genetic test results with.

Who Discussed Results with (Select All That Apply)	*N*	Percent
Family Member(s)	223	88.1
Friend(s)	213	84.2
Veterinarian not employed by Wisdom Panel	138	54.5
Co-workers/Colleagues	118	46.6
Contacts on social networking services (e.g., Facebook)	74	29.2
Other Animal-Related Professional	35	13.8
Wisdom Panel Customer Service	8	3.2
Wisdom Panel Veterinarian/Genetic Counsellor	3	1.2
Genetic Specialist/Counsellor not employed by Wisdom Panel	1	0.4
Other:	-	-
Generalized groups of people (e.g., people meet at dog park)	4	1.6
Agency where acquired dog (breeder, shelter)	3	1.2
Someone they re-homed the dog with	2	0.8
Prospective Adopters	1	0.4
Veterinarian (Not specified if Wisdom Panel associate or not)	1	0.4
“Good Dog”	1	0.4

**Table 7 animals-12-03360-t007:** Who participant would consult with if they had a question about any aspects of their dog’s genetic test results.

Person(s) Would Consult with if Had a Question Regarding Dog’s Genetic Test Results (Select All That Apply)	*N*	Percent
A veterinary care provider	194	76.7
Wisdom Panel customer service	87	34.4
Wisdom Panel veterinarian/genetic counselor	75	29.6
Health and medical websites	43	17.0
Spouse/Significant other	26	10.3
I would not consult with anyone	23	9.1
Other family member(s)	21	8.3
[Dog’s name]’s trainer	20	7.9
Friend(s)	16	6.3
The organization I adopted [dog’s name] from	11	4.3
Groups on social media like Facebook or Reddit	8	3.2
[Dog’s name]’s groomer	8	3.2
The breeder and/or store I purchased [dog’s name] from	5	2.0
Other		
“Own research”/“Online research”	3	1.2
Rescue Groups who Specialize in the “Primary Breeds”	1	0.4

**Table 8 animals-12-03360-t008:** Additional context to consumer behavioral responses to their dog’s canine genetic test results.

How Results Were Implemented or Other Responses (Select All That Apply)	*N*	Percent
I did nothing with the test results.	171	67.6
I altered [dog’s name]’s training.	53	20.9
I altered the ways in which I care for [dog’s name] such as their diet.	24	9.5
I altered [dog’s name]’s medical care.	13	5.1
Other	-	-
Used Results for Behavior/Personality Insights	5	2.0
Gave Results to Veterinarian	3	1.2
No Changes Needed/No Insights Given	3	1.2
Other-Unrelated	3	1.2
Have Not Received Results	3	1.2
Satisfied Curiosity/Peace of Mind	2	0.8
Advocated Against “Designer Dogs”	1	0.4
Used to Guide Future Care	1	0.4
Did further research into breed traits and behavior	1	0.4
Weight-Loss Plan	1	0.4

## Data Availability

The link to the data will be made available after acceptance. https://doi.org/10.5281/zenodo.6558199.

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
