# Peer review of "A Study Evaluating Consumer Motivations, Perceptions, and Responses to Direct-to-Consumer Canine Genetic Test Results"

_animals, 2022, doi:10.3390/ani12233360_

Round 1

Reviewer 1 Report

Congratulations to the authors on this research study.  I think this paper has great potential to contribute to the literature if the authors can make the justification/need for the study clearer and more compelling.  To me, the Methods and Results were very easy to follow.  My main feedback is in the Introduction and Discussion sections.  Somewhere in the Introduction the authors need to speak to the importance of this work-  why is this research needed, beyond just being interesting?  What are the implications of this line of research?   In the Discussion, I think quite a bit more time should be spent interpreting and discussing the findings of the current study, including talking about why consumers care so much about knowing their dog’s breed (Why are we so obsessed with breeds?  Why do breeds matter to identity and social status?  What drives these beliefs?).  I also feel more time should be spent discussing the implications of this research for human- non-human animal relationships.

Detailed feedback is listed by section below.  I hope it is helpful to the authors:

Abstract

-       On lines 27-28 the authors state the ‘majority’ of participants perceived the results as accurate (52%) and that the service provides great value (49.6%).  52% is barely a majority and 49.6% is not a majority. This is a very quick fix, but I suggest the authors re-word to reflect the finding more accurately.

Introduction

-       Sentence beginning on line 65, : Another study by Lewis and Mellersh [8] was the first to empirically evaluate the efficacy of DTC-GTs – something that has been called into question– by using genetic information collected by the UK Kennel Club.  Clarity needed-  this study evaluated the efficacy of DTC-GTs to do what? The following sentence needs clarity as well: Their results found meaningful declines in harmful mutations, implying DTC-GTs can have positive benefits on animal health.  How do they have benefits on animal health by finding harmful mutations?  

-       Sentence beginning line 71 is also confusing: Subsequently, they found associations between breed: health (i.e., morbidity rates), levels of inbreeding, and morphology (e.g., brachycephaly.

-       I’m not clear on what the authors mean when they say social science research on this topic has been essential for ‘improving the use and implementation of such tests’ (line 76-77).  Could the authors clarify what the mean here?  Maybe breaking the existing sentence into 2 parts ( ‘improving the use and implementation of such tests’ and ‘providing insight into motivations, understanding, behavioral responses’).

-       After reading the Introduction, I am unclear on the implications of this research.  Will addressing the study objectives improve human-companion animal relations?  Companion animal quality of life?  If so, how?  I think the authors should add a bit more to justify the need for this research.

Methods

-       Methods are clear and easy to follow!

Results

-       Delete template info lines 184-186, otherwise Results clear and easy to follow

Discussion

            -Why do consumers want to know the ‘background’ on their dogs so badly?  Are certain breeds considered desirable?  Tough?  Scary?  The best?  The worst?  I think it is incredibly interesting that consumers might use these tests to embrace/confirm their identity, but I was dying to hear the authors tell me more about how or why the dog’s breed could have such an impact on identity. Please explain more on how breed might affect identity and social status!  

            -If the authors hypothesis is that the use of DTC-GT could affect dog guardians social identity, why did they not ask about this in the survey to explore whether this hypothesis is supported?  Is this worth exploring in future research?  It is not mentioned as a future direction… 

            -On line 353 the authors point out that human sector markets focus more on ancestry than health results, while the pet industry markets health over ancestry.  I am unclear on why the authors think this might be the case and was looking for a hypothesis of some sort. While I don’t know about the pet tests, the focus on the ancestry data for the human tests is likely because it has long been considered much more accurate than the health results (and there are major legal implications if health info is wrong/hard to interpret/distressing/etc).  I’m not sure if there are similar discrepancies between the ancestry and health data accuracy for the pet tests, but I assume so if the technology is the same.  This all seems like something highly relevant to discuss or at least mention in this paper…

-To me, the paragraph on cloning and bringing back extinct dog breeds as well as the paragraph on privacy concerns felt out of place in the Discussion.  Unless the authors can more clearly explain the relevance of these topics to the current paper theme/research questions, I would suggest removing. 

            -In the Limitations & Future directions paragraph, the authors spend time explaining details of how this particular survey could be improved if used again.  That is fine, but I am more interested in studies that will build off these studies based on the knowledge gained here.  Could the authors speak more to those future directions?

Conclusions

            -In the opening sentence, the authors state that understanding X, X and X are essential ‘to evaluate the potential impacts (both direct and indirect) this industry may have’.  What direct and indirect impacts are the authors referring to?  This is very unclear to me.

            -The closing sentence stresses that the factors studied in this paper will continue to be important areas of research focus.  I am stilling missing why the authors feel this is such an important area-  its all very interesting, but what are the implications for people and ‘pets’?

Author Response

Reviewer 1

Congratulations to the authors on this research study.  I think this paper has great potential to contribute to the literature if the authors can make the justification/need for the study clearer and more compelling.  To me, the Methods and Results were very easy to follow.  My main feedback is in the Introduction and Discussion sections.  Somewhere in the Introduction the authors need to speak to the importance of this work-  why is this research needed, beyond just being interesting?  What are the implications of this line of research?   In the Discussion, I think quite a bit more time should be spent interpreting and discussing the findings of the current study, including talking about why consumers care so much about knowing their dog’s breed (Why are we so obsessed with breeds?  Why do breeds matter to identity and social status?  What drives these beliefs?).  I also feel more time should be spent discussing the implications of this research for human- non-human animal relationships.

Thank you for your kind and invaluable feedback to improve this manuscript. As discussed further below, we have incorporated all comments into revising this manuscript. Your time in reviewing and providing detailed, insightful feedback is highly appreciated.

On lines 27-28 the authors state the ‘majority’ of participants perceived the results as accurate (52%) and that the service provides great value (49.6%).  52% is barely a majority and 49.6% is not a majority. This is a very quick fix, but I suggest the authors re-word to reflect the finding more accurately.

We had condensed the inclusion of both “strongly agree” and “somewhat agree” responses due to the word count. We have re-added the “somewhat agree” responses to support the characterization of participant responses.

Sentence beginning on line 65, : Another study by Lewis and Mellersh [8] was the first to empirically evaluate the efficacy of DTC-GTs – something that has been called into question– by using genetic information collected by the UK Kennel Club.  Clarity needed-  this study evaluated the efficacy of DTC-GTs to do what? The following sentence needs clarity as well: Their results found meaningful declines in harmful mutations, implying DTC-GTs can have positive benefits on animal health.  How do they have benefits on animal health by finding harmful mutations? 

Sentence beginning line 71 is also confusing: Subsequently, they found associations between breed: health (i.e., morbidity rates), levels of inbreeding, and morphology (e.g., brachycephaly.

We have used both of these comments to revise this section to communicate these ideas more clearly. This section now reads:

Another study by Lewis and Mellersh [8] empirically evaluated the efficacy of DTC-GTs – something that has been called into question (e.g., [9]) –to determine if allele frequency for autosomal recessive diseases changed following DNA tests becoming commercially available. Their results found meaningful declines in harmful mutation frequencies in UK Kennel Club registered dog populations, implying DTC-GTs can have positive benefits on animal health when incorporated into breeding practices. This trend is particularly salient to purebred dog populations as they are more likely to have health problems. For example, Bannasch et al. [10] used genomic data derived from DTC tests to make comparisons between breed standards and breed-specific health outcomes. Subsequently, they found associations between certain dog breeds (e.g., brachycephalic dogs) having higher morbidity rates and higher levels of inbreeding.

I’m not clear on what the authors mean when they say social science research on this topic has been essential for ‘improving the use and implementation of such tests’ (line 76-77).  Could the authors clarify what the mean here?  Maybe breaking the existing sentence into 2 parts ( ‘improving the use and implementation of such tests’ and ‘providing insight into motivations, understanding, behavioral responses’).

We have revised this sentence to now say:

Social science research into human self-use of DTC-GTs has been essential to improving the industry by providing insight into consumer motivations to use genetic services, their understandings of test results, and behavioral responses to their genetic test results (e.g., [11, 12, 13]).

After reading the Introduction, I am unclear on the implications of this research.  Will addressing the study objectives improve human-companion animal relations?  Companion animal quality of life?  If so, how?  I think the authors should add a bit more to justify the need for this research.

We have added to the last paragraph to provide insight into the implications of this research that are discussed further in the discussion. This now reads:

To overcome the limitations of passive data use (i.e., Amazon Reviews, company website data), we collaborated with Wisdom PanelTM to recruit participants  from their existing consumer database. By collecting data directly from consumers, our goal was to further evaluate their motivations, perceptions, and responses to their using DTC DNA services for their dog. These areas of research are important to understand consumer experiences and potential implications to the human-dog bond. As such, our objectives were to:....

Methods are clear and easy to follow!

Thank you for this comment.

Delete template info lines 184-186, otherwise Results clear and easy to follow

Thank you for catching this. We have now removed those lines.

Why do consumers want to know the ‘background’ on their dogs so badly?  Are certain breeds considered desirable?  Tough?  Scary?  The best?  The worst?  I think it is incredibly interesting that consumers might use these tests to embrace/confirm their identity, but I was dying to hear the authors tell me more about how or why the dog’s breed could have such an impact on identity. Please explain more on how breed might affect identity and social status! 

We have revised the discussion to include these considerations about societal influences on breed identity and have added to the framework for future consideration.

If the authors hypothesis is that the use of DTC-GT could affect dog guardians social identity, why did they not ask about this in the survey to explore whether this hypothesis is supported?  Is this worth exploring in future research?  It is not mentioned as a future direction…

This was not a hypothesis and is worth considering for future research. We have added this to the discussion in both the previously mentioned revision and the future directions section.

On line 353 the authors point out that human sector markets focus more on ancestry than health results, while the pet industry markets health over ancestry.  I am unclear on why the authors think this might be the case and was looking for a hypothesis of some sort. While I don’t know about the pet tests, the focus on the ancestry data for the human tests is likely because it has long been considered much more accurate than the health results (and there are major legal implications if health info is wrong/hard to interpret/distressing/etc).  I’m not sure if there are similar discrepancies between the ancestry and health data accuracy for the pet tests, but I assume so if the technology is the same.  This all seems like something highly relevant to discuss or at least mention in this paper…

This determination is based off citation [14] as the author’s reported this pattern of companion animal DTC-GT companies marketing health information as the primary benefit, with the breed results second. We have removed the sentence about the human sector as it no longer applied with other revisions to the discussion.

As to the point about test accuracy, we have clarified this more in the introduction based on it needing more clarity about test efficacy being questioned by other scholars.

To me, the paragraph on cloning and bringing back extinct dog breeds as well as the paragraph on privacy concerns felt out of place in the Discussion.  Unless the authors can more clearly explain the relevance of these topics to the current paper theme/research questions, I would suggest removing.

 We have removed this from the discussion and placed it in the conclusion as it relates to our concluding remarks on intended/unintended implications of the market and associated technologies.

In the Limitations & Future directions paragraph, the authors spend time explaining details of how this particular survey could be improved if used again.  That is fine, but I am more interested in studies that will build off these studies based on the knowledge gained here.  Could the authors speak more to those future directions?

Thank you for this suggestion and we have now added future directions past improving the survey to also complement the importance of this research and our discussion on breed identity.

In the opening sentence, the authors state that understanding X, X and X are essential ‘to evaluate the potential impacts (both direct and indirect) this industry may have’.  What direct and indirect impacts are the authors referring to?  This is very unclear to me.

We have added 2 sentences immediately following the referenced statement to clarify what is meant by (in)direct impacts of the industry to say:

Understanding pet guardian motivations, perceptions, and reactions to genetic test results are essential to evaluate potential impacts (both direct and indirect) this industry may have. Direct implications to consider are those intended by the industry, such as improving animal health outcomes as previously found [10], while indirect impacts are those not specifically advertised by pet animal DTC-GT companies. Examples to consider are consumer responses to unwanted test results (e.g., negative breed stereotypes) or this technology being used for other purposes.

The closing sentence stresses that the factors studied in this paper will continue to be important areas of research focus.  I am stilling missing why the authors feel this is such an important area-  its all very interesting, but what are the implications for people and ‘pets’?

Thank you for your time in reviewing and providing very helpful feedback about this manuscript. We have added to both the discussion and conclusion to provide more information about the importance of this research and to consider potential human-animal bond influences.

Reviewer 2 Report

This paper examines a timely, original topic, and one I have wondered about myself. Just when I wondered whether anyone was doing research on dog DNA tests, the invitation to review this manuscript arrived in my in-box. It is a well conducted study, comprehensively grounded in the literature. It is well organized and well written. The analysis is clear. The conclusion suggests avenues for additional research. 

Author Response

Reviewer 2

This paper examines a timely, original topic, and one I have wondered about myself. Just when I wondered whether anyone was doing research on dog DNA tests, the invitation to review this manuscript arrived in my in-box. It is a well conducted study, comprehensively grounded in the literature. It is well organized and well written. The analysis is clear. The conclusion suggests avenues for additional research.

Thank you for your time in reviewing and providing feedback on our manuscript. We are also grateful for your comments and appreciation of the topic.

Reviewer 3 Report

The authors take up an interesting issue. Genetic research on humans and animals is increasing of interest not only to scientists but also to people not related to science. Dog owners pay more and more attention to the health, origin and behaviour of their animals.

In my opinion, the work can be published in Animals with minor adjustments. I share my comments below.

line 90-95: in my opinion, it would be better to place this fragment already in the material and methods subchapter, more precisely in 2.1. Survey Measures and Procedures

line 127: was this the complete end of the survey for these respondents? It would be worth asking them why. Is it a price issue? No interest in the topic? But this is a note for the future, for further research.

132-135: very good idea with using the name of the dog, psychologically encourages the respondent - the guardian of the dog - to complete the survey to the end

I would add in the text the information that the entire questionnaire was placed as supplementary files.

line 168: I'm not convinced that a description of software should be considered a reference. In my opinion, it is enough to provide the name and version of the program, without reference to the bibliography and then to the website.

line 184-186: in my opinion this part is redundant. Dividing the results section into subsections is standard.

line 269: remove "returned"

In the work, the authors refer to 50 bibliographic sources. In 14 of them, the Internet is listed as the source. In my opinion, quite a few of these online items.

Author Response

Reviewer 3

line 90-95: in my opinion, it would be better to place this fragment already in the material and methods subchapter, more precisely in 2.1. Survey Measures and Procedures

We have made this change in which we moved the operationalization of “dog guardian” to section 2.1 and have changed this paragraph to highlight the importance of this study as recommended by reviewer 1 and to include the study’s objectives.

line 127: was this the complete end of the survey for these respondents? It would be worth asking them why. Is it a price issue? No interest in the topic? But this is a note for the future, for further research. 

We are currently planning a follow-up study to look at this population more generally, as compared to one consumer database. We will ask those who have not used a test about their reasons for not doing so. For this study, because those who received the recruitment email should have only been a customer based on Wisdom Panel’s participant criteria, we did not collect information from them as it was likely that the person who received the email was not the one who was the guardian of the dog (e.g., purchased as a gift).

132-135: very good idea with using the name of the dog, psychologically encourages the respondent - the guardian of the dog - to complete the survey to the end

Thank you for this comment.

I would add in the text the information that the entire questionnaire was placed as supplementary files.

At the end of section 2.1, we have added the following sentence: “Survey questions are provided as a supplementary file.”

line 168: I'm not convinced that a description of software should be considered a reference. In my opinion, it is enough to provide the name and version of the program, without reference to the bibliography and then to the website.

We have removed the formal citations for SPSS and Microsoft Excel and added SPSS version 28 to the text.

line 184-186: in my opinion this part is redundant. Dividing the results section into subsections is standard.

These lines were removed as it was an error. The text was from Animals’ formatting and should not have been included. Thank you for catching this.

line 269: remove "returned"

“returned” has been removed from the 3.2 section subheader.

In the work, the authors refer to 50 bibliographic sources. In 14 of them, the Internet is listed as the source. In my opinion, quite a few of these online items.

We believe the full comment here was not transferred over so we will attempt to address this as best as possible. Most of the internet sources are our referencing company websites directly as there are no other sources to support this information. For example, we felt it was important to show that other companies in this market or those associated with it have implications, such as genome banking services or current veterinary models to include genetic test services. The other sources labeled as internet are to indicate the article is only available online such as the [45] citation. Based on editor feedback and formatting the references to Animals’ standards, we will likely adjust the reference citation style. If we have missed something from this comment that needs to be further addressed, please let us know and we will follow-up.